# Association between Fathers’ Use of Heated Tobacco Products and Urinary Cotinine Concentrations in Their Spouses and Children

**DOI:** 10.3390/ijerph19106275

**Published:** 2022-05-21

**Authors:** Ayumi Onoue, Yohei Inaba, Kentaro Machida, Takuya Samukawa, Hiromasa Inoue, Hajime Kurosawa, Hiromitsu Ogata, Naoki Kunugita, Hisamitsu Omori

**Affiliations:** 1Department of Biomedical Laboratory Sciences, Faculty of Life Sciences, Kumamoto University, 4-24-1 Kuhonji, Chuo-ku, Kumamoto 862-0976, Japan; ayumi.onoue@gmail.com; 2Department of Environmental Health, National Institute of Public Health, Minami, Wako-shi 351-0197, Japan; inaba.y.aa@niph.go.jp; 3Department of Pulmonary Medicine, Graduate School of Medical and Dental Sciences, Kagoshima University, Kagoshima 890-8520, Japan; machida@m.kufm.kagoshima-u.ac.jp (K.M.); inoue@m2.kufm.kagoshima-u.ac.jp (H.I.); 4Ikeda Hospital, Kanoya 893-0024, Japan; samukawa07@ikeda-hp.com; 5Department of Occupational Health, Tohoku University Graduate School of Medicine, Sendai 980-8575, Japan; kurosawa-thk@m.tohoku.ac.jp; 6Graduate School of Nutrition Sciences, Kagawa Nutrition University, 3-9-21, Sakado 350-0288, Japan; ogata.hiromitsu@eiyo.ac.jp; 7School of Health Sciences, University of Occupational and Environmental Health, Kitakyushu 807-8555, Japan; kunugita@med.uoeh-u.ac.jp

**Keywords:** heated tobacco products, secondhand smoke, cotinine, biomarker

## Abstract

Heated tobacco products (HTPs) have become increasingly popular among smokers, especially among young adults in Japan in recent years. Assessments of secondhand tobacco smoke (SHS) exposure due to HTPs are scarce. The present study aimed to assess the urinary levels of total nicotine metabolites (TNMs) of non-smoking spouses and their children following SHS exposure due to their fathers’ use of HTPs. A total of 41 families including 129 participants were recruited between 2018 and 2021. The number of non-smoking spouses and children of the fathers who smoke combustion cigarettes, the fathers who use HTPs, and the fathers who are non-users or have never smoked was 27, 66, and 36, respectively. The urinary levels of TNMs, including cotinine (Cot) and 3′-hydroxycotinine (3-OHCot), were measured using liquid chromatography/tandem mass spectrometry (LC/MS/MS). The spouses and children of fathers who use HTPs had significantly higher levels of TNMs in their urine compared to those with fathers who were non-smokers or non-users. The current study is the first to assess SHS exposure due to HTP use, and to suggest the importance of strategies to prevent exposure to SHS from HTP use in public places and educational strategies to protect non-smokers from secondhand HTP aerosol exposure in households and other private places.

## 1. Introduction

Heated tobacco products (HTPs) are tobacco products that produce aerosols by heating tobacco in battery-powered heating systems at lower temperatures than in conventional tobacco-burning cigarettes (generally < 600 °C) [1,2]. HTPs contain nicotine, which is a highly addictive substance, and other chemicals that are inhaled by users through the mouth [1,2]. HTPs have been sold in Japan since November 2014. Tabuchi et al. reported that 4.7% of the population used HTPs or e-cigarette products in 2017 [3]. HTPs have become increasingly popular among smokers, especially among young adults in Japan in recent years. The JASTIS study estimated the prevalence of HTP use to be 11.3% among the entire Japanese population, and over 30% among current cigarette smokers in 2019 [4]. Data from the National Health and Nutritional Examination Survey conducted by the Ministry of Health, Labour, and Welfare, revealed that the prevalence of HTP use was 26.7% among current smokers in 2019 [5].

The Framework Convention on Tobacco Control (WHO FCTC) effectively protects the public from the harmful effects of tobacco smoke [6]. HTPs are subject to the provisions of the WHO FCTC. In contrast, partial smoking bans are ineffective [7,8]. The implementation of comprehensive smoke-free policies by the WHO FCTC is the only way to effectively protect the public from the harms of SHS. New regulations from the revised Health Promotion Law in Japan allow the use of HTPs in areas where people eat and drink [8,9]. 

There is currently no evidence indicating that HTPs are less harmful than conventional tobacco products [1,2]. Furthermore, there is insufficient evidence regarding the potential health effects of secondhand emissions produced by HTPs, though the emissions are known to contain some harmful and potentially harmful constituents [1,2].

HTPs contain nicotine, a highly addictive substance found in tobacco which, in turn, makes HTPs addictive [1,2]. The concentrations of nicotine in tobacco filters and mainstream HTP smoke are similar to those in conventional combustion cigarettes [10]. The biomonitoring of urine nicotine metabolites has been used extensively to assess the extent of tobacco smoke exposure [11] and is suitable for the assessment of secondhand tobacco smoke (SHS) exposure [9,11,12]. It is highly predictive of adverse health outcomes among children [13,14,15]. Accurate and reliable measurements of HTP exposure are essential for identifying and confirming the patterns of HTP use and for assessing their potential biological effects in human populations [14,16].

SHS exposure among people living with a combustion cigarette smoker is currently a popular topic [17,18,19]. However, assessments of SHS exposure due to HTP use are scarce. Therefore, the present study was designed to assess the urinary levels of total nicotine metabolites (TNMs) of non-smoking spouses and their children following SHS exposure due to their fathers’ use of HTPs.

## 2. Materials and Methods

### 2.1. Study Design and Participants

The present cross-sectional study was conducted between April 2018 and March 2021 in Japan. Participants were recruited from the following regions in Japan: Kumamoto, Kagoshima, and Miyagi Prefecture. Men who worked in and around these regions, their spouses and their children were recruited for the study. In the Kumamoto region, workers were recruited through the Kumamoto research and study group for occupational health nursing. Participants included families with spouses and children under 20 years of age. In this study, we excluded fathers who smoked combustion cigarettes and those who co-used HTPs. We also excluded families in which both the fathers and spouses smoked.

A total of 129 participants were enrolled. The study population consisted of nine fathers who smoked combustion cigarettes, their non-smoking spouses (*n* = 9), and their non-smoking children (*n* = 18); 22 fathers who used heated tobacco products (HTPs), their non-smoking spouses (*n* = 22), and their non-smoking children (*n* = 44); and 10 fathers who have never smoked combustion cigarettes and were non-HTP users, their non-smoking spouses (*n* = 10), and their non-smoking children (*n* = 26). In the present study, we defined current combustion cigarette smokers as participants who only smoke cigarettes daily, and current HTP users as participants who only use HTPs daily, and have done so for at least 2 weeks.

For the main analysis, we used an analysis of variance. An ANOVA with a significance level of 5%, a power of 0.8, and an effect size of 0.4 resulted in a sample size of 21 [20]. We required a sample size of over 21 in each of the three groups.

The study was conducted in accordance with the Declaration of Helsinki, and was approved by the Institutional Review Board (or Ethics Committee) of Kumamoto University (protocol code 1510). All subjects provided written informed consent to participate in the present study.

### 2.2. Measurement

#### 2.2.1. Questionnaire for SHS Exposure and Definitions

SHS exposure was assessed using a self-report questionnaire (Table 1). The core questionnaire contained information on demographics, smoking behavior, and exposure to SHS. Data were obtained at the time of urine sample collection. The questions included in the questionnaire were: (1) “Have you smoked during the time with your spouse?” (yes/no) for the father; (2) “Have you smoked during the time with your children?” (yes/no) for the father; (3) “Has your husband smoked during the time with you?” (yes/no) for the spouse, and (4) “Has your father smoked during the time with you?” (yes/no) for the children.

If the participant answered “yes” in response to questions (1) or (3), their spouses were classified as having been exposed to SHS. If the participant answered “yes” in response to questions (2) or (4), their children were classified as having been exposed to SHS.

#### 2.2.2. Determination of Urinary Total Nicotine Metabolites

All spouses and children provided the first-morning urine samples. One urine sample was collected from each participant. The samples were stored at −20 °C. The urinary levels of TNMs, including those of cotinine (Cot) and 3′-hydroxycotinine (3-OHCot), were measured primarily based on the results of liquid chromatography [LC]/mass spectrometry [MS]/[MS] (LC-MS/MS). This was performed in the Department of Environmental Health in the National Institute of Public Health, which is a member of the WHO Collaborating Centers for tobacco control.

Urine creatinine-corrected values were calculated in this study because this correction helps to avoid the confounding effects of overly diluted or hyper-concentrated urine [14]. The results were expressed as cotinine/creatinine ratios in ng/mg creatinine. [21]

#### 2.2.3. LC/MS/MS System

##### Apparatus

A liquid chromatography/tandem mass spectrometry (LC/MS/MS) system (Waters Corporation, Milford, MA, USA) was used with an ultra-performance LC (Acquity series) and a mass spectrometer (VEVO TQ-S).

Data were acquired and processed using MassLynx version 4.1 software (Waters Corporation, Milford, MA, USA). A CORTECS UPLC HILIC analytical column with a particle size of 1.6 μm and an i.d. of 100 mm × 2 mm (Waters Corporation, Milford, MA, USA) was also used. Solution A and solution B for the mobile phase mixture comprised 100 mmol/L ammonium formate/0.03% formic acid (10/90 *v/v*) and acetonitrile, respectively. HPLC elution was conducted using 20% A for 1 min, followed by a linear gradient from 20% A to 100% B for 50 min, and maintenance for 10 min. The flow rate of the mobile phase was 0.3 mL/min, the column temperature was 40 °C, and the injection volume was 2 μL.

Mass spectrometric detection was conducted in positive ESI mode. The mass spectrometric parameters were optimized for the following protonated ions: *m/z* 177.2→98.0 [cotinine], *m/z* 180.3→80.0 [cotinine-d3], *m/z* 193.2→80.0 [3-hydroxycotinine], and *m/z* 196.3→80.0 to attain maximum sensitivity. The MS conditions were set as follows: nitrogen desolvation gas, 1000 L/h; nitrogen cone gas, 150 L/h; source temperature, 150 °C; desolvation temperature, 550 °C; capillary voltage, 2.4 kV; and cone voltage, 30 V. Cotinine and 3-hydroxycotinine determination was set at 0.005 ng/mL. Calibration curves for cotinine and 3-hydroxycotinine were prepared in the 0.005–10 ng/mL range.

##### Sample Preparation

A 0.25 mL urine sample was mixed with 0.75 mL of water, 2 mL of buffer (pH 4.5), 0.25 mL of I.S. (100 ng/mL cotinine-*d_3_* and 3-hydroxycotinine-*d_3_*), and 0.065 mL of 2N NaOH. The mixture was added to an activated ENVI–Carb solution (250 mg/6 mL), which was washed with 2 mL of water followed by 3 mL of 20% methanol. Cotinine and 3-hydroxycotinine were eluted with acetonitrile (2.5 mL). The elution was then transferred to the sample vials.

##### Creatinine Assay

Urinary creatinine levels were determined using the LabAssay Creatinine assay (Wako, Japan). The kit was mixed with diluted urine and reagents using the Jaffe reaction. Urinary creatinine levels were quantified by measuring the absorbance at 520 nm using a microplate reader (SUNRISE) from TECAN (Tecan Group Ltd., Männedorf, Switzerland).

### 2.3. Ethical Considerations

All subjects provided written informed consent to participate in the present study, which was conducted in accordance with the Declaration of Helsinki and the Ethical Guidelines for Epidemiological Research (partially revised on 1 December 2008 by the Ministry of Education, Culture, Sports, Science, and Technology and the Ministry of Health, Labour, and Welfare). The human ethics committee of Kumamoto University approved the research protocol (no. 1510).

### 2.4. Data Analysis

Data are presented as the number of cases (*n*) (percentage), means (standard deviation: SD), or means (standard error: SE). The normality of the distribution was assessed using the Shapiro-Wilk test. One-way analysis of variance (ANOVA) and the Kruskal-Wallis test were used to assess the differences in characteristics, and the urinary levels of total nicotine metabolites (TNMs), namely cotinine and 3′-hydroxycotinine, among the three groups: only combustion cigarette smokers, only HTP users, and never-smokers and non-users. The post hoc Scheffe’s test was used to assess the differences in the urinary levels of TNM among the 3 groups. The chi-square test was used to assess t categorical variables. Statistical significance was set at *p* < 0.05. All statistical analyses were conducted using IBM SPSS Statistics software (version 27.0, IBM Japan, Tokyo, Japan). There were no missing data in the present study.

## 3. Results

### 3.1. Study Population Characteristics

Table 2 summarizes the characteristics associated with the SHS exposure status from fathers. This study included a total of 41 families (129 participants). The number of non-smoking spouses and children of the fathers who smoke combustion cigarettes, non-smoking spouses and children of the fathers who use HTPs, and non-smoking spouses and children of the fathers who are non-users or have never smoked was 27, 66, and 36, respectively. The study population consisted of 41 non-smoking spouses and 88 non-smoking children. The numbers of non-smoking spouses and non-smoking children of the fathers who only smoke combustion cigarettes, fathers who only use HTPs, and fathers who are never-smokers and non-users were 9 and 18, 22 and 44, and 10 and 26, respectively. The mean ages of the participants did not differ significantly among the three groups.

### 3.2. SHS Exposure in Spouses and Children from Fathers Defined by Urinary Levels of TNMs

Table 3 shows the urinary levels of TNMs after creatine normalization according to the SHS exposure status from fathers. Figure 1 shows the comparison of TNM urinary levels after creatine normalization among the three groups (total non-smoking spouses and children, *n* = 129). In this study, the non-smoking spouses and children of the fathers who smoke combustion cigarettes (*n* = 27) had an average urinary TNM concentration of 0.0107 nmol/mg creatinine (SE = 0.0021). The non-smoking spouses and children of the fathers who use HTPs (*n* = 66) had an average urinary TNM concentration of 0.0058 nmol/mg creatinine (SE = 0.0011). The non-smoking spouses and children of the fathers who are non-users or never-smokers (*n* = 36) had an average urinary TNM concentration of 0.0012 nmol/mg creatinine (SE = 0.0003). The urinary TNM concentrations among the non-smoking spouses and children of the fathers who smoked combustion cigarettes were significantly higher than the urinary TNM concentrations among the families in which the fathers were never-smokers or non-users (Table 3 and Figure 1). The urinary TNM concentrations among HTP users tended to be lower than those among combustion cigarette smokers. The urinary TNM concentrations in the non-smoking spouses and children of the fathers who use HTPs were significantly higher than those among the non-smoking spouses and children of the fathers who were never-smokers or non-users (Table 3 and Figure 1).

In this study, the percentage of spouses and children exposed to SHS in the HTP group was higher than in the combustion cigarette group (Table 3). The urinary TNM concentrations for the children of the fathers who answered “yes” to questions indicating SHS exposure from HTPs were significantly higher than for those with fathers who were never-smokers or non-users (Table 3).

## 4. Discussion

To the best of our knowledge, the current study was the first to assess SHS exposure due to HTP use. We assessed the association between SHS exposure (due to the fathers’ use of HTPs) and urinary TNM concentrations (obtained using LC-MS/MS assays) in non-smoking spouses and children.

We found that urinary TNM concentrations in the spouses and children of fathers who use HTPs were significantly higher, especially in the group who answered “yes” to questions indicating SHS exposure, than for those whose fathers were never-smokers and non-users.

The biomonitoring of urine cotinine levels has been used to assess SHS exposure [11]. In the current study, urinary levels of TNMs, including cotinine (Cot) and 3′-hydroxycotinine (3-OHCot), were measured. Cotinine is metabolized to 3-OHCot, which is the most abundant nicotine metabolite and accounts for 38% of all urinary metabolites in humans [22,23]. It is well known that cotinine levels are highly dependent on the duration of time since the last cigarette was smoked. The half-lives of cotinine and hydroxycotinine are 6–22 h and 4.6–8.3 h, respectively [24,25,26]. The sum of cotinine (Cot) and 3′-hydroxycotinine (3-OHCot) is a useful biomarker for estimating daily nicotine intake [27]. In this study, the internal estimated daily nicotine intake could not be calculated. Further studies are required to estimate the daily nicotine intake for this population. Based on the results of the urinary levels of TNMs, 4-(methylnitrosamino)-1-(3-pyridyl)-1-butanol (NNAL), which has a longer half-life than TNMs, may also be a suitable biomarker for SHS exposure. Further research on NNAL is necessary.

In the current study, the percentage of spouses and children exposed to SHS in the HTP group was higher than in the combustion cigarette group. A recent study by Imura et al. in Japan reported that 32.6% of individuals aged 15–19 years were exposed to secondhand HTP aerosols [28]. Gravely et al. reported that nearly half of all smokers believed HTPs to be less harmful than combustion cigarettes [29]. Furthermore, HTP users were significantly more likely to believe that HTPs are less harmful than cigarettes compared to non-HTP users [29]. Therefore, the fathers who used HTPs in this study were likely to believe that HTPs could be used safely during the time they spend with their spouses and children. Ohmomo and their colleagues observed that HTP users (who switched from using combustible tobacco <2 years prior) displayed abnormal DNA methylation and transcriptome profiles, albeit to a lesser extent than those who used combustible tobacco [30]. Long-term molecular epidemiologic studies are needed.

Several studies suggest that secondhand HTP aerosol exposure may have harmful side effects [10,31,32,33]. The levels of some harmful substances were lower in HTPs than in combustible cigarettes, however, several substances were higher [33]. A recent study found that non-smoking women (aged 40 years or older) who were exposed to ETS due to their husbands’ combustion cigarette smoking had the lowest FEV_1_/FVC [34]. Another study reported that cumulative exposure to SHS at work may contribute to the deterioration of pulmonary function in non-smoking employees [35]. Imura et al. reported that the incidence of asthma attacks and chest pain due to HTP aerosol exposure was higher than that caused by cigarette smoke, suggesting that secondhand HTP aerosol exposure may lead to respiratory and cardiovascular abnormalities [28]. These findings suggest that cumulative exposure to secondhand HTP aerosols may contribute to the deterioration of pulmonary function in non-smoking spouses and children. Large-scale prospective studies are required to confirm these findings.

Igarashi et al. reported that younger or more affluent people tend to use HTPs [36]. Our findings strongly suggest that educational strategies are also necessary to protect non-smokers from secondhand HTP aerosol exposure in households and other private places.

New regulations from the revised Health Promotion Law in Japan stipulate that the use of HTPs is allowed in areas where people eat and drink [8,9]. The present results suggest that secondhand HTP aerosol exposure may be associated with an increased risk of health deterioration. Therefore, the Health Promotion Law in Japan should be revised again to prohibit the use of HTPs in environments where aerosol exposure is likely such as public spaces (e.g., restaurants).

To the best of our knowledge, no previous studies have described an association between SHS exposure due to HTP use and urinary cotinine concentrations. The strengths of the current study lie in the quantification of individual levels of SHS exposure using the LC/MS/MS system. The urinary levels of TNMs, including those of cotinine (Cot) and 3′-hydroxycotinine (3-OHCot), were measured mainly based on LC-MS/MS in the Department of Environmental Health in the National Institute of Public Health, which is a member of the WHO Collaborating Centers for tobacco control. To the best of our knowledge, no previous studies have described a link between a families’ exposure to SHS due to HTP use and urinary levels of TNMs.

The present study had several limitations. First, the number of participants enrolled in this cross-sectional study was relatively small. This was of particular concern when analyzing the sample according to participants’ SHS exposure status, which may involve weak statistical precision. In this study, the non-smoking spouses and children of the fathers who smoke combustion cigarettes (*n* = 27) had an average urinary TNM of 2.0 ng/mg creatinine (SD = 2.0). The non-smoking spouses and children of the fathers who use HTPs (*n* = 66) had an average urinary TNM concentration of 1.1 ng/mg creatinine (SD = 1.62). The non-smoking spouses and children of the fathers who are non-users or have never smoked (*n* = 36) had an average urinary TNM concentration of 0.23 ng/mg creatinine (SD = 2.16) (data not shown). Assuming a pooled SD of approximately 2.0, this study has over 80% statistical power for the ANOVA (analysis of variance) with a 5% significance level. Thus, the target sample size is estimated to be sufficient for the analysis [20].

For the t-test of the mean difference between the non-smoking spouses and children of the fathers who use HTPs and the non-smoking spouses and children of the fathers who are non-users or have never smoked, the study had about 90% statistical power with a 5% significance level [20]. Thus, the target sample size was estimated to be sufficient for the total analysis.

Second, since self-reported SHS exposure was used, participants’ recall and reporting bias regarding HTP use could not be disregarded. In this study, the question of “How many times have you smoked in front of your spouse or children?” was not included. However, self-reported SHS exposure correlates well with biomarker concentrations [37]. Third, we could not define other sources of SHS exposure (e.g., spending time with other smokers). In the present study, the mean age of the children was 9.1 years, suggesting that they are preschool or school-age children. School-age children appear to spend a lot of time at home [18].

Fourth, information regarding the last exposure before sample collection was missing. However, all spouses and children provided the first-morning urine samples. Cotinine (Cot) is the major metabolite of nicotine, and its longer half-life (16–18 h) makes it a good biomarker for nicotine uptake [14]. The measurement of urine TNM levels, including those of cotinine (Cot) and 3′-hydroxycotinine (3-OHCot), is believed to provide a reliable index for recent nicotine exposure (over the past several days) [38]. Differences in the duration of time since the last exposure may not influence the differences observed among three groups. Further studies related to this point are necessary.

Fifth, the internal estimated daily nicotine intake could not be calculated. Further studies are needed to estimate the daily intake of nicotine in this population. Sixth, in this study, information regarding participants’ socioeconomic status was missing. Socioeconomic status may be related to the type of consumption and the prevalence of SHS. Further studies are needed to explore this point.

Despite these limitations, these data may contribute to the development of evidence-based recommendations for more effective smoke-free laws to protect non-smokers from secondhand HTP aerosol exposure. Large-scale prospective studies are needed to confirm the potential health deterioration risks associated with HTPs.

## 5. Conclusions

The urinary cotinine concentrations of the spouses and children of the fathers who use HTPs were significantly higher compared to those with fathers who were non-users. To our knowledge, this is the first study to examine families’ exposure to SHS due to HTP use and to suggest the importance of strategies to prevent exposure to SHS from HTPs. The findings of this study indicate that public health policymakers should both enact and enforce a more comprehensive smoking ban for HTP use in public places, and also implement educational strategies to protect non-smokers from secondhand HTP aerosol exposure in households and other private places.

## Figures and Tables

**Figure 1 ijerph-19-06275-f001:**
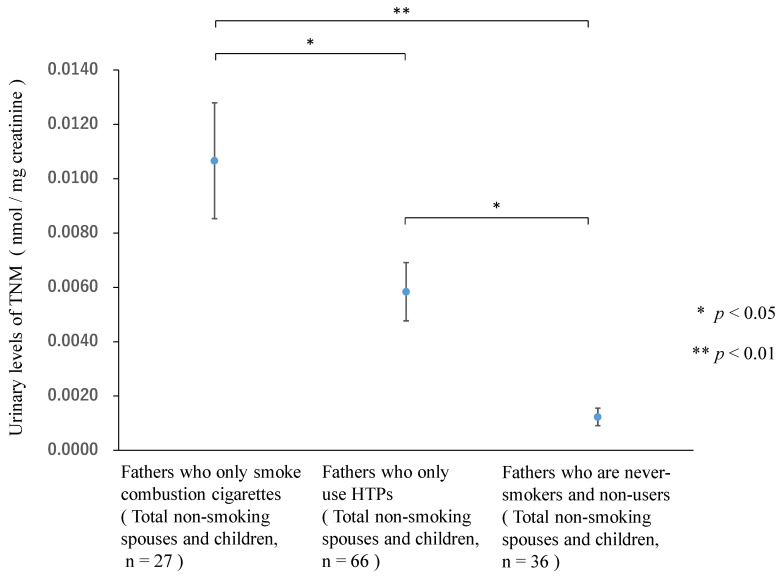
Comparison of total nicotine metabolite (TNM) urinary levels after creatine normalization among the three groups. (conc. Creatine).

**Table 1 ijerph-19-06275-t001:** A self-report questionnaire for SHS exposure and definitions of SHS.

(1) “Have you smoked during the time with your spouse?” (yes/no) for the father
(2) “Have you smoked during the time with your children?” (yes/no) for the father
(3) “Has your husband smoked during the time with you?” (yes/no) for the spouse
(4) “Has your father smoked during the time with you?” (yes/no) for the children
Definition of SHS exposure
For spouses, if the participant answered “yes” in response to questions (1) or (3), their spouses were classified as being exposed to SHS.
For children, if the participant answered “yes” in response to questions (2) or (4), their children were classified as being exposed to SHS.

Abbreviations: SHS, secondhand smoke.

**Table 2 ijerph-19-06275-t002:** The characteristics of the study participants.

		SHS Exposure Status from Fathers			
Characteristics	Total Participants (41 Families)	Fathers Who Only Smoke Combustion Cigarettes (9 Families)	Fathers Who Only Use HTPs (22 Families)	Fathers Who Are Never-Smokers and Non-Users (10 Families)	*p*-Value
Total non-smoking spouses and children	*n* = 129	*n* = 27	*n* = 66	*n* = 36	
Age, years, M (SD)	18.1 (14.1)	18.8 (14.5)	18.3 (14.3)	17.3 (13.5)	0.93
Male, *n* (%)	43	7	23	13	
Female, *n* (%)	86	20	43	23	0.28
Non-smoking spouses	*n* = 41	*n* = 9	*n* = 22	*n* = 10	
Age, years, M (SD)	37.6 (6.0)	37.7 (7.5)	37.5 (6.3)	37.7 (4.4)	0.99
Female, *n* (%)	41 (100)	9 (100)	22(100)	10 (100)	
Non-smoking children	*n* = 88	*n* = 18	*n* = 44	*n* = 26	
Age, years, M (SD)	9.1 (4.4)	9.3 (4.8)	8.8 (3.9)	9.4 (4.9)	0.83
Male, *n* (%)	43 (48.9)	7 (38.9)	23 (52.3)	13 (50.0)	
Female, *n* (%)	45 (51.1)	11 (61.1)	21 (47.7)	13 (50.0)	

Notes: Data are expressed as means (standard deviation), or as number (*n*) (percentage). Abbreviations: HTPs, heated tobacco products.

**Table 3 ijerph-19-06275-t003:** The urinary levels of TNMs after creatine normalization according to the SHS exposure status from fathers.

		SHS Exposure Status from Fathers			
Characteristics	Total Participants (41 Families)	Fathers Who Only Smoke Combustion Cigarettes, (9 Families)	Fathers Who Only Use HTPs, (22 Families)	Fathers Who Are Never-Smokers and Non-Users, (10 Families)	*p*-Value
Total non-smoking spouses and children TNM, nmol/mg creatinine, M (SE)	*n* = 129	*n* = 27	*n* = 66	*n* = 36	
		0.0107 (0.0021) **	0.0058 (0.0011) *	0.0012 (0.0003)	<0.001
SHS exposure, Yes		*n* = 15	*n* = 49	*n* = 0	
		0.0107 (0.0025) **	0.0063 (0.0014) *		<0.001
SHS exposure, No		*n* = 12	*n* = 17	*n* = 36	
		0.0106 (0.0038) **	0.0045 (0.0015)	0.0012 (0.0003)	<0.001
Non-smoking spouses TNM, nmol/mg creatinine, M (SE)	*n* = 41	*n* = 9	*n* = 22	*n* = 10	
		0.0083 (0.0035) *	0.0027 (0.0005)	0.0010 (0.0004)	0.01
SHS exposure, Yes		*n* = 6	*n* = 19	*n* = 0	
		0.0087 (0.0052) *	0.0029 (0.0006)		0.028
SHS exposure, No		*n* = 3	*n* = 3	*n* = 10	
		0.0074 (0.0034) **	0.0018 (0.0009)	0.0010 (0.0004)	0.008
Non-smoking children TNM, nmol/mg creatinine, M (SE)	*n* = 88	*n* = 18	*n* = 44	*n* = 26	
		0.0119 (0.0027) **	0.0074 (0.0015) *	0.0013 (0.0004)	<0.001
SHS exposure, Yes		*n* = 9	*n* = 30	*n* = 0	
		0.0121 (0.0024) **	0.0084 (0.0021) *		0.001
SHS exposure, No		*n* = 9	*n* = 14	*n* = 26	
		0.0116 (0.0050) **	0.0051 (0.0017)	0.0013 (0.0004)	0.003

Notes: Data are expressed as means (standard error). Abbreviations: SHS, secondhand smoke; HTPs, heated tobacco products; TNMs, total nicotine metabolites; SE, standard error. * *p* < 0.05, ** *p* < 0.01 compared with fathers who were never-smokers and non-users.

## Data Availability

Not applicable.

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
