# Peer review of "Association between Fathers’ Use of Heated Tobacco Products and Urinary Cotinine Concentrations in Their Spouses and Children"

_ijerph, 2022, doi:10.3390/ijerph19106275_

Round 1
Reviewer 1 Report
The study entitled “Association between Fathers’ Use of Heated Tobacco Products and Urinary Cotinine Concentrations in Their Spouses and Children” is conducted to compare the exposure to secondhand environmental tobacco smoke (ETS) of subjects whose father or spouse is smoking heating tobacco products (HTPs) with those who are living with a combustion cigarette or non-smoking father or spouse.
The study has merits. The topic is important to be discussed. Currently, we don’t have many numbers of specific studies to aim the exposure to ETS through HTPs. Objectives are clear. The research methods including both conducting the self-reported survey and urinary tobacco exposure biomarkers analysis are done well. The selection of subjects was done appropriately. Results are also analyzed well with appropriate statistical analysis.
I have a few comments as below, which I hope authors may find useful.
1. Study on the exposure to secondhand tobacco smoke (SHS) for people who are living in the same family as a smoker is a popular topic these days. I feel that the authors may be able to present a better literature review in the introduction part. I can suggest some recently published works, sorted by publication date, as drops from an ocean (In 2022: https://doi.org/10.3390/ijerph19063746; in 2021: https://doi.org/10.1038/s41598-021-84017-y; in 2020: https://doi.org/10.1038/s41598-020-66920-y). Authors are encouraged to consider discussing such studies and highlight the significance of their conducted works.
2. Urinary total nicotine metabolites (TNM) as biomarkers can be an appropriate approach. However, it heavily depends on the amount of exposed nicotine. If applicable, the authors are invited to consider calculating the internal estimated daily intake for nicotine and conduct the comparison for the subject groups. In this case, authors can refer to these published works: https://doi.org/10.3390/ijerph19063746, or https://doi.org/10.1093/ntr/ntz034. Particularly, by doing this we can specify and assess the intake of tobacco from combustion cigarettes and HTPs.
3. The 4-(methylnitrosamino)-1-(3-pyridyl)-1-butanol (NNAL) can also be a suitable biomarker to be tracked for SHS exposure (Please see https://doi.org/10.1038/s41598-021-84017-y). It may be useful to the readers if the authors may explain the reason for selecting urinary TNM as the tracking biomarker.
It might be useful to the readers if the authors may wish to discuss the reason that they have considered urinary TNM as the tracking biomarker.
4. Since the molecular weights of cotinine and hydroxycotinine are different, it may be more reasonable to use the molarity composition unit (e.g. nmol/mg-creatinine) for TNM.
5. Please consider providing an appropriate caption for the y-axis in figure 1. Showing only a unit can be confusing for some readers. Also, please confirm that in figure 1, *<0.05 means p<0.05.
Author Response
Answer for Reviewer 1 ijerph-1694199 Onoue A et al.
Comments and Suggestions for Authors
The study entitled “Association between Fathers’ Use of Heated Tobacco Products and Urinary Cotinine Concentrations in Their Spouses and Children” is conducted to compare the exposure to secondhand environmental tobacco smoke (ETS) of subjects whose father or spouse is smoking heating tobacco products (HTPs) with those who are living with a combustion cigarette or non-smoking father or spouse.
The study has merits. The topic is important to be discussed. Currently, we don’t have many numbers of specific studies to aim the exposure to ETS through HTPs. Objectives are clear. The research methods including both conducting the self-reported survey and urinary tobacco exposure biomarkers analysis are done well. The selection of subjects was done appropriately. Results are also analyzed well with appropriate statistical analysis.
I have a few comments as below, which I hope authors may find useful.
We really appreciate your important comments. According to your comments, we amended and added the sentences as below. Onoue A et al.
[ Comments and Answers]
Comment 1. Study on the exposure to secondhand tobacco smoke (SHS) for people who are living in the same family as a smoker is a popular topic these days. I feel that the authors may be able to present a better literature review in the introduction part. I can suggest some recently published works, sorted by publication date, as drops from an ocean (In 2022: https://doi.org/10.3390/ijerph19063746; in 2021: https://doi.org/10.1038/s41598-021-84017-y; in 2020: https://doi.org/10.1038/s41598-020-66920-y). Authors are encouraged to consider discussing such studies and highlight the significance of their conducted works.
Answer 1: Thank you for your comments.
According to your comments, we added the sentence below in the Introduction session「SHS exposure in people living in the same family with a combustion-cigarette smoker is currently a popular topic [17-19].」, and added three references [17-19].
Comment 2. Urinary total nicotine metabolites (TNM) as biomarkers can be an appropriate approach. However, it heavily depends on the amount of exposed nicotine. If applicable, the authors are invited to consider calculating the internal estimated daily intake for nicotine and conduct the comparison for the subject groups. In this case, authors can refer to these published works: https://doi.org/10.3390/ijerph19063746, or https://doi.org/10.1093/ntr/ntz034. Particularly, by doing this we can specify and assess the intake of tobacco from combustion cigarettes and HTPs.
Answer 2: Thank you for your comments.
According to your comments, we added the sentence below in the discussion session as limitation. 「In this study, the internal estimated daily intake for nicotine could not be calculated. Further studies are required to estimate daily intake for nicotine. 」
Comment 3. The 4-(methylnitrosamino)-1-(3-pyridyl)-1-butanol (NNAL) can also be a suitable biomarker to be tracked for SHS exposure (Please see https://doi.org/10.1038/s41598-021-84017-y). It may be useful to the readers if the authors may explain the reason for selecting urinary TNM as the tracking biomarker.
It might be useful to the readers if the authors may wish to discuss the reason that they have considered urinary TNM as the tracking biomarker.
Answer 3: Thank you for your comments.
We added the sentence below in discussion session.
「Based on the results of urine cotinine levels of TNMs, the 4-(methylnitrosamino)-1-(3-pyridyl)-1-butanol (NNAL), of which half-life is longer than TNMs, can also be a suitable biomarker to be tracked for SHS exposure. Further research on NNAL is necessary.」
Comment 4. Since the molecular weights of cotinine and hydroxycotinine are different, it may be more reasonable to use the molarity composition unit (e.g. nmol/mg-creatinine) for TNM.
Answer 4: Thank you for your important comments.
We calculated “ nmol/mg-creatinine “ and changed “ nmol/mg-creatinine “ from “ ng/mg-creatinine ” in the Table 3 and Figure 1.
Comment 5. Please consider providing an appropriate caption for the y-axis in figure 1. Showing only a unit can be confusing for some readers. Also, please confirm that in figure 1, *<0.05 means p<0.05.
Answer 5: Thank you for your comments.
We changed caption for the y-axis and *p<0.05, **p<0.01 in figure 1

Reviewer 2 Report
The authors address a relevant question regarding the exposure to second hand smoking due to heated tobacco products. Conducting this kind of research is particularly relevant, since as the authors state, the public may think that these procuts are harmless. The results are very interesting, particularly the fact, that among HTP-user there was more second hand smoking.
Regarding the conduction of the study there are some issues which need to be addressed before publication.
First: The authors shuold describe the way they recruited the participants and where. In addition, inclusion and exclusion criteria need to be reported.
Second: Regarding data collection, did they collect any other data? E.g. socioeconomic status may be related to the type of consumption and the prevalence of SHS, thus this would be relevant.
Third: There is a lack of information regarding the last exposure before taking the sample. This should have been collected and needs to be reported, since any differences in the timepoint of the last exposure may explain differences between groups.
Finally: The questions used to address SHS are very vague. Maybe this is a problem of translation. The authors should clarify this. I would think they mean somehow "Have you ever smoked while being whith your spouse?". It would have been better to ask with a time horizont, ot to ask how many times, etc. ("In the past n (e.g. two) weeks, did you smoke in front of your children?" "How many times..."). If the authors did not asked in detail, this should be discussed as a limitation.
The authors did not consider other sources of exposure to SHS (e.g. spending time with other smokers (not the own husband/parent). Did they address this? A brief statement about this issue should be included in the discussion.
Regarding the presentation of results: The text in lines 160-172 repeats very much of the information of table 2. The authors should try to avoid redundancy and highlight the most important facts. In general the Tables and Figure have a very small typo difficult to read. Please use bigger letters.
Author Response
Answer for Reviewer 2 ijerph-1694199 Onoue A et al.
Comments and Suggestions for Authors
The authors address a relevant question regarding the exposure to second hand smoking due to heated tobacco products. Conducting this kind of research is particularly relevant, since as the authors state, the public may think that these procuts are harmless. The results are very interesting, particularly the fact, that among HTP-user there was more second hand smoking.
Regarding the conduction of the study there are some issues which need to be addressed before publication.
We really appreciate your important comments. According to your comments, we amended and added the sentences as below. Onoue A et al.
[ Comments and Answers]
First: The authors shuold describe the way they recruited the participants and where. In addition, inclusion and exclusion criteria need to be reported.
Answer: Thank you for your important comments.
According to your comments, we added the sentence below in the Materials and Methods session. 「Participants were recruited from the following regions in Japan: Kumamoto, Kagoshima and Miyagi prefecture. Subjects included families with spouses and children under 20 years of age. In this study, we excluded fathers who smoke combustion cigarette and co-use HTPs. We also excluded the family if both the father and spouse were smokers.」
Second: Regarding data collection, did they collect any other data? E.g. socioeconomic status may be related to the type of consumption and the prevalence of SHS, thus this would be relevant.
Answer: Thank you for your comments.
According to your comments, we added the sentence below in the discussion session as limitation「In this study, Information regarding the socioeconomic status was missing. Socioeconomic status may be related to the type of consumption and the prevalence of SHS. Further studies are needed in this point.」
Third: There is a lack of information regarding the last exposure before taking the sample. This should have been collected and needs to be reported, since any differences in the timepoint of the last exposure may explain differences between groups.
Answer: Thank you for your comments.
According to your comments, we added the sentence below in the discussion session as limitation.「Fourth, all spouses and children provided the first-morning urine samples. However, information regarding the last exposure before sample collection was missing. Differences in the last exposure times may explain differences in results among the three groups. 」
Finally: The questions used to address SHS are very vague. Maybe this is a problem of translation. The authors should clarify this. I would think they mean somehow "Have you ever smoked while being whith your spouse?". It would have been better to ask with a time horizont, ot to ask how many times, etc. ("In the past n (e.g. two) weeks, did you smoke in front of your children?" "How many times..."). If the authors did not asked in detail, this should be discussed as a limitation.
The authors did not consider other sources of exposure to SHS (e.g. spending time with other smokers (not the own husband/parent). Did they address this? A brief statement about this issue should be included in the discussion.
Answer: Thank you for your important comments.
According to your comments, we added the sentence below in the discussion session as limitation.「Second, since self-reported SHS exposure was used, recall and reporting bias of THP use could not be disregarded. In this study, the question of "How many times have you smoked in front of your spouse or children?" was not included. However, self-reported SHS exposure correlates well with biomarker concentrations [37]. Third, we could not define other sources of SHS exposure (e.g., spending time with other smokers). In the present study, mean age of children was 9.1, suggesting that they are preschool or school-age children. School-age children appear to spend a lot of time at home [18]. 」
Regarding the presentation of results: The text in lines 160-172 repeats very much of the information of table 2. The authors should try to avoid redundancy and highlight the most important facts. In general the Tables and Figure have a very small typo difficult to read. Please use bigger letters.
Answer: Thank you for your important comments.
According to your comments, we added the sentence below in the Results.
「Table 2 summarizes the characteristics associated with SHS exposure status from fathers. This study included a total of 41 families (129 participants). The numbers of the non-smoking spouses and children of fathers who smoke combustion cigarettes, the non-smoking spouses and children of fathers who use HTPs, and the non-smoking spouses and children of fathers who are never-smokers and non-users were 27, 66 and 36, respectively. The study population consisted of 41 non-smoking spouses and 88 non-smoking children. The numbers of non-smoking spouses and non-smoking children of fathers who only smoke combustion cigarettes, fathers who only use HTPs, and fathers who are never-smokers and non-users were 9 and 18, 22 and 44, and 10 and 26, respectively. The mean ages of the participants did not differ significantly among the three groups.」
We used bigger letters in Tables and Figure.

Round 2
Reviewer 1 Report
I have checked the authors' responses as well as the revised manuscript. Overall, the manuscript has been improved, and responses are reasonable. This is an interesting approach, and I wish to read more from the authors in the future.
Author Response
Round 2 Answer for Reviewer 1 ijerph-1694199 Onoue A et al.
[ Comments and Answers]
I have checked the authors' responses as well as the revised manuscript. Overall, the manuscript has been improved, and responses are reasonable. This is an interesting approach, and I wish to read more from the authors in the future.
Answer: We really appreciate your important comments.
Onoue A et al.

Reviewer 2 Report
The atuhors have tried to address my points in the first review.
Regarding recruitment it is still not mentioned how they recruited, the authors have added now the locations but not the method of recruitment.
I consider that not asking the timepoint of last exposure is a shortcoming. The authors have written something about it in the discussion, but they do not provide a real discussion on it: "However, information regarding the last exposure before sample collection was missing. Differences in the last exposure times may explain differences in results among the three groups.". I would expect that the authors discuss more deeply whether this is a problem or not, do they think the differences are due to differences in last time exposure or do they consider that in their study this is not likely.
The other points have been addressed satisfactorily.
Author Response
Round 2 Answer for Reviewer 2 ijerph-1694199 Onoue A et al.
[ Comments and Answers]
Comment 1: The authors have tried to address my points in the first review.
Regarding recruitment it is still not mentioned how they recruited, the authors have added now the locations but not the method of recruitment.
Answer 1 : Thank you for your important comments.
According to your comments, we added and amended the sentence below in the "2.1. Study Design and Participants" in the Materials and Methods session.
「The present cross-sectional study was conducted between April 2018 and March 2021 in Japan. Participants were recruited from the following regions in Japan: Kumamoto, Kagoshima, and Miyagi prefecture. Men who worked at the company from in and around the regions and their spouses and children were recruited for the study. In kumamoto region, workers were recruited through the kumamoto research and study group of occupational health nursing. Participants included families with spouses and children under 20 years of age. In this study, we excluded fathers who smoke combustion cigarette and those who co-used HTPs. We also excluded the families in which both the fathers and spouses were smokers.」
Comment 2: I consider that not asking the timepoint of last exposure is a shortcoming. The authors have written something about it in the discussion, but they do not provide a real discussion on it: "However, information regarding the last exposure before sample collection was missing. Differences in the last exposure times may explain differences in results among the three groups.". I would expect that the authors discuss more deeply whether this is a problem or not, do they think the differences are due to differences in last time exposure or do they consider that in their study this is not likely.
Answer2 : Thank you for your important comments.
According to your comments, we added and amended the sentence below in the "2.1. Study Design and Participants" in the Materials and Methods session.
「Fourth, information regarding the last exposure before sample collection was missing. However, all spouses and children provided the first-morning urine samples. Cotinine (Cot) is the major metabolite of nicotine, and its longer half-lives (16-18 h) makes it a good biomarker for nicotine uptake [14]. Measurement of urine TNMs levels, including those of cotinine (Cot) and 3’-hydroxycotinine (3-OHCot), is believed to provide a reliable index of recent (past several days) nicotine exposure [38]. Differences in last time exposure may not influence the differences among three groups. Further studies related to this point are necessary.
The other points have been addressed satisfactorily.
We appreciate for reviewing our manuscript.
